# Preliminary Evaluation of an Advanced Ventilation-Control Algorithm to Optimise Microclimate in a Commercial Broiler House

**DOI:** 10.3390/ani14233430

**Published:** 2024-11-27

**Authors:** Kehinde Favour Daniel, Lak-yeong Choi, Se-yeon Lee, Chae-rin Lee, Ji-yeon Park, Jinseon Park, Se-woon Hong

**Affiliations:** 1Department of Rural and Bio-Systems Engineering, Chonnam National University, Gwangju 61186, Republic of Korea; kenniedee@jnu.ac.kr (K.F.D.); cly6847@jnu.ac.kr (L.-y.C.); seyeonn@jnu.ac.kr (S.-y.L.); chaerinl@jnu.ac.kr (C.-r.L.); qkrwldus7749@jnu.ac.kr (J.-y.P.); 2Education and Research Unit for Climate-Smart Reclaimed-Tideland Agriculture, Chonnam National University, Gwangju 61186, Republic of Korea; icarus381@jnu.ac.kr; 3AgriBio Institute of Climate Change Management, Chonnam National University, Gwangju 61186, Republic of Korea

**Keywords:** ventilation control, ventilation rate requirement, energy balance, cooling pads, heat stress reduction, broiler performance

## Abstract

Maintaining a good environment for broiler chickens is essential for their health, growth, and welfare. This study developed and tested a new ventilation control algorithm designed to improve the indoor temperature in mechanically ventilated broiler houses. The algorithm calculates the right amount of ventilation needed to maintain optimal conditions. We compared the performance of the new algorithm with the current ventilation control system in a commercial broiler farm. During a high-temperature period, the new algorithm helped reduce indoor temperatures by 1.5 to 2 °C, which led to less heat stress for the chickens. Although the use of cooling pads increased, the electric energy savings from the reduced use of ventilation fans were significant. The new algorithm also reduced chicken mortality by 16.5%, which indicates that it could improve their overall welfare and productivity. This approach could help farmers use energy more efficiently while creating better bird conditions.

## 1. Introduction

As of 2022, the livestock products market in South Korea was valued at approximately 37.5 trillion Korean won, which reflects a positive growth trend over the past five years [1]. Large-scale facilities that utilise intensive broiler farm operations are increasingly being established to meet the rising demand [2]. As a result, efficient microclimate control has become essential [3,4,5]. Proper ventilation plays a critical role in maintaining optimal indoor temperatures within broiler houses. In addition, poor air circulation can lead to the accumulation of harmful gases, such as ammonia and carbon dioxide (CO_2_), which can adversely affect bird welfare, growth rate, and overall productivity [3,4,5,6,7,8,9,10].

Sustaining an acceptable microclimate in a broiler farm can be challenging, particularly in countries such as South Korea, which are subject to significant seasonal temperature variations [11]. In Korea, the air temperature varies significantly between extreme weather periods. For example, in January (the coldest month), the monthly mean temperatures range between −6 and 7 °C, while in August (the warmest month), the monthly mean temperatures range from 23 to 27 °C, with a maximum of 34 °C [12]. In broiler houses, the effects of weather events are more pronounced depending on the birds’ age. Usually, at the later rearing stage, maximum cooling system capacity is needed [13,14,15].

The most cost-effective way to control the microclimate, including temperature, humidity, and gases, in a large-scale broiler house is through ventilation. In a mechanically ventilated system, the ventilation rate needed for microclimate regulation determines the number and operating status of the ventilation fans. Therefore, it is crucial to determine the appropriate ventilation rate at every moment, as either excessive or insufficient ventilation can lead to failure in maintaining optimal microclimate conditions.

Traditionally, ventilation rates in broiler houses are calculated using static formulas based on bird age, weight, and environmental factors [16]. However, these methods often fail to account for the dynamic nature of the broiler house environment and the complex interactions between various factors affecting indoor temperature [17]. For these reasons, several studies were conducted to develop dynamic control strategies for ventilation based on various influencing factors. Using a dynamic thermal environment simulation, Zhang et al. [18] tested three control strategies: either-or temperature control, either-or-neither temperature control, and temperature–humidity control (THC). The group reported that THC was superior to temperature-only controls. Gates [19] developed an automated ventilation control system based on static pressure and temperature control and tested its operation in a broiler research room for two years. Detsch et al. [20] developed a real-time semi-physical model based on optimisation- and learning algorithms to control the temperature inside broiler houses and showed the potential to maintain optimal temperatures better than the current control system.

However, despite these efforts to improve ventilation control, many commercial broiler houses still rely on the operator’s experience for effective ventilation management. This is because the ventilation control methods that have been developed have not been widely evaluated or demonstrated in actual broiler house settings. Therefore, this study aims to develop a ventilation control algorithm that can improve the indoor microclimate of commercial broiler houses by accurately estimating the required ventilation rate.

More specifically, this was performed as follows: (1) The required ventilation rate was calculated based on heat-balance analysis for commercial broiler houses and compared with the currently applied ventilation rate. (2) The developed ventilation control algorithm was implemented in a commercial broiler house to compare its microclimate control performance with the existing ventilation control algorithm. In addition, because the algorithm developed in this study is still in the pilot phase, it currently only addresses indoor temperature as a control factor, but there are plans to expand to humidity and gas conditions in the future.

## 2. Materials and Methods

### 2.1. Experimental Farm

The experiments were carried out in a mechanically ventilated broiler house located in Namwon City, the southern region of South Korea. The experimental farm was selected because the region is one of the top regions for broiler farming in South Korea. The broiler house comprised a floor area of 87 × 14 m^2^, a ridge height of 5 m, and an eave height of 3.5 m, housing 30,000 broiler chickens. A tunnel ventilation system consisted of 14 tunnel fans, each with a capacity of 37,000 m^3^ h^−1^, two cooling pads, six circulation fans, and 58 sidewall air inlets (29 on each side), as shown in Figure 1. All ventilation facilities and automatic control systems were established using the Koko-farm system (Emotion Co., Ltd., Jeonju-si, Republic of Korea). The experimental farm comprises two identical broiler houses. Building 1 used the existing microclimate control system, while Building 2 was operated using the newly developed ventilation control algorithm developed in this study.

### 2.2. Data Acquisition

In the existing system, temperature and humidity were measured at four points in the longitudinal direction of both buildings (at a height of 0.7 m above the floor) at one-minute intervals (LO-1000, Koko-farm system, Emotion Co., Ltd., Jeonju-si, Republic of Korea) as shown in Figure 1. The average values from the four points were used to calculate the indoor temperature and humidity for ventilation control. The microclimate control system was operated using a programmable logic controller (PLC) to collect data and control the ventilation actuators. The operation of the tunnel fans and cooling pads was recorded at five-minute intervals from the PLC. Furthermore, the outdoor temperature and humidity were measured at the same intervals as the indoor parameters by a weather station installed on the experimental farm. The weight of the broilers was recorded in real-time using six IoT scales (Flat Use IoT Scale, Koko-farm system, Emotion Co., Ltd., Republic of Korea) that were evenly distributed across the floor of the broiler house—see Figure 1. These measurements were used to predict the average weight of the broilers.

In the newly developed ventilation control algorithm, indoor temperature and humidity were recorded at 10 s intervals, and the operation signals of the tunnel fans and cooling pads were recorded in JSON (JavaScript Object Notation) format by the PLC. The JSON files were then converted to match the same 10 s intervals as the indoor data using MATLAB R2021a (Mathworks Inc., Natick, MA, USA).

### 2.3. Ventilation Control Algorithms

The current ventilation control algorithm used in broiler houses controls the operation of tunnel fans based on indoor temperature. The operation of tunnel fans comprises six fan stages. Stage 0 represents the minimum ventilation stage, during which two tunnel fans operate intermittently using an on/off system. The minimum ventilation rates were set as 1233 m^3^ h^−1^ at the beginning of the rearing period and increased with broiler age, reaching 22,611 m^3^ h^−1^ at harvest. In Stages 1 through 5, four, six, eight, ten, and fourteen tunnel fans operate continuously, respectively. In the current system, the selection of the appropriate fan stage is determined based on indoor temperatures. The operator programs the system with the start temperature for each fan stage based on the optimal growth temperature for chickens at different ages. This is referred to as the set temperature, which is generally determined empirically by the operator. In the experimental farm, the set temperature for each fan stage for different chicken ages was configured as shown in Figure 2. When the indoor temperature of the broiler house reaches the specified temperature, the corresponding fan stage is activated. The indoor temperature is updated every 10 s via the PLC, and accordingly, the fan stage is also updated every 10 s. The cooling pads’ operation is controlled similarly, based on a threshold indoor temperature set by the operator for each chicken age. When the indoor temperature reaches the threshold, the cooling pad is activated.

The new ventilation control algorithm proposed in this study aims to regulate ventilation by calculating the ventilation rate requirement (VRR) based on the heat energy balance inside the broiler house. This approach is expected to offer three key advantages: First, it eliminates the need for the operator’s empirical settings to operate tunnel fans and cooling pads. Second, instead of relying solely on indoor temperature to operate tunnel fans, it considers the difference between indoor and outdoor temperatures, using the resulting heat exchange to control ventilation. Third, it incorporates the sensible and latent heat produced by the chickens, as well as heat exchange through the building envelope, into the calculation of the VRR. In both the current and new algorithms, the air inlets are automatically controlled to maintain a negative indoor pressure level between 22.4 and 37.4 Pa. Therefore, the algorithms do not control the opening and closing of the air inlets.

The VRR is first determined by calculating the amount of heat energy that needs to be removed from the building through ventilation—see Equation (1). This calculation includes the energy needed for cooling from the indoor temperature to the set temperature, the sensible and latent heat produced by the chickens, and the energy transferred through the building envelope. Then, using Equation (2), the VRR is calculated by dividing the energy to be removed through ventilation by the building’s heat capacity and the temperature difference between indoor and outdoor air (which includes the cooling effect of the cooling pads). As shown in Equation (3), the cooling pad efficiency is defined as the ratio of the actual temperature drop achieved by the cooling pads to the difference between the dry-bulb and wet-bulb temperatures of the outside air. When the cooling pad is not operating, the efficiency is set to zero.
(1)$$Q_{v}=\rho C_{p}V_{b}\frac{T_{i}-T_{d}}{∆t}+Q_{a}+Q_{l}+Q_{d},$$
(2)$$VRR=\frac{Q_{v}}{\rho C_{p}(T_{i}-T_{o}-T_{c})},$$
(3)$$T_{c}=(T_{o}-T_{ow})\times \varepsilon$$

Here, $Q_{v}$ is the heat loss requirement due to the ventilation (W); ρ denotes the air density (kg m^−3^); $C_{p}$ is the specific heat of the air (J kg^−1^ °C^−1^); $V_{b}$ signifies the volume of the broiler house (m^3^); $T_{i}$ is the indoor temperature (°C); $T_{d}$ is the optimal growth temperature or set temperature (°C); ∆t denotes the control time and was set to 150 s; $Q_{a}$ is the sensible heat generation of the chicken (W); $Q_{l}$ denotes the latent heat generation of the chicken (W); $Q_{d}$ is the heat transfer through the walls of the building (W); VRR is the ventilation rate requirement (m^3^ s^−1^); $T_{o}$ represents the outdoor temperature (°C); $T_{c}$ is the temperature drop due to the cooling pads (°C); $T_{ow}$ is the outdoor wet-bulb temperature (°C); and ε denotes the cooling efficiency of the cooling pads, estimated as 0.65 based on measurements [22].

The sensible heat produced by the chickens was estimated using an empirical formula based on indoor temperature and the chicken’s body weight [23,24]. The latent heat was estimated by assuming that 40% of the daily water intake of the chicken evaporates inside the building, which contributes to evaporative latent heat [24]. To account for the variability in the chicken’s metabolic activity over 24 h, a sine function was used to model fluctuations for both sensible and latent heat production [25,26]. The mass of individual broilers was estimated using data on weekly broiler weights provided by the National Institute of Animal Science in Korea [27], as described in Equation (7).
(4)$$Q_{a}=m^{0.75}(307.87-15.63T_{i}+0.3105T_{i}^{2})\times \frac{4184}{24\times 3600}\times S_{c},$$
(5)$$Q_{l}=-0.4\times \left(583.5-17.1T_{i}+25.25m\right)\times \frac{1}{24\times 3600}\times S_{c},$$
(6)$$S_{c}=1-a\;\sin\left(\frac{2\pi }{24}\times \left(h+6-h_{min}\right)\right),$$
(7)$$m=-35.783+19.098d+0.6008d^{2}$$

Here, m is the mass of the chicken (kg); $S_{c}$ denotes the sine function for the variability in the chicken’s activity over 24 h; a is the constant, 0.21 for $Q_{a}$ and 0.46 for $Q_{l}$; h represents the time of day in hours; $h_{min}$ is the constant, 0.38 for $Q_{a}$ and 0.67 for $Q_{l}$; and d signifies the age of the chicken in days.

The heat transfer through the building envelope was calculated using the thermal transmittance (U-value) and the surface area of the walls as follows:(8)$$Q_{c}=-\left(U_{w}A_{w}+U_{f}A_{f}\right)\left(T_{i}-T_{o}\right)$$

Here, $U_{w}$ is the thermal transmittance or U-value of the wall and was found to be 0.247 W m^−2^ K^−1^ through measurements [22]; $U_{f}$ denotes the thermal transmittance of the floor and was estimated as 2.75 W m^−2^ K^−1^ for a typical concrete slab floor; $A_{w}$ represents the surface area of the wall (m^2^); and $A_{f}$ is the floor area (m^2^).

In the control algorithm, VRR is divided by the ventilation capacity per fan to determine the number of fans that need to operate. It then determines the appropriate fan stage. However, a numerical error may occur when the denominator in Equation (2) approaches zero due to a very small temperature difference between indoor and outdoor air, which can cause the VRR to become excessively large. Moreover, when the indoor temperature is close to the set point, there is a risk of mechanical wear due to the frequent changes in the fan stage. Frequent operation of the cooling pads can also cause significant problems for farms with limited water supply. To prevent these problems, exception clauses were added to the control algorithm to handle such cases:

Exception 1: if $Q_{v}<0$ or $T_{i}<T_{d}+T_{da}$, the fan stage is 0

Exception 2: if $T_{o}\geq T_{d}+T_{da}\times 3$ and $T_{i}>T_{d}+T_{da}\times 2+0.5\;{}^{\circ}C$, the cooling pads are operated until the indoor temperature decreases to $T_{i}\leq T_{d}+T_{da}\times 2$.

Exception 3: if $T_{o}\geq T_{d}+T_{da}$, the fan stage is calculated based on $\min\left(VRR,\;{VRR}_{2}\right)$.

Here, $T_{da}$ is the allowable temperature range and determined as 1.5 °C; and VRR_2_ is defined as the ventilation rate required to completely exchange the indoor air once during a period of ∆t (${VRR}_{2}=V_{b}/∆t$).

The three exception clauses were applied in numerical order, and when an exception was met, the subsequent clauses were ignored. Exception 1 was applied when the indoor temperature was lower than the set point, which would lead to fan stage 0, i.e., operating at a minimum ventilation rate. The set temperature was defined up to 1.5 °C ($T_{da}$) below the optimal temperature. Exception 2 applied if Exception 1 was not met and the outdoor temperature was more than 4.5 °C higher than the set temperature. In this case, the cooling pads were operated when the indoor temperature exceeded the set temperature by 3.5 °C. It stopped once the indoor temperature dropped to within 3 °C of the set point. If the operation and stopping conditions were the same, the cooling pads would continuously cycle between operation and stop, ranging from 3.5 °C to 3.6 °C. Therefore, different conditions were set for operation and stopping. Subsequently, the VRR and fan stage were recalculated based on the reduced temperature of the incoming air due to the cooling pads. Exception 3 applied if neither Exception 1 nor 2 was met. In other words, the cooling pads were not operated, and the outdoor temperature was higher than the indoor temperature. In this case, since there may not be a significant difference between the indoor and outdoor temperatures, the VRR calculated using Equation (2) could be excessively high. Therefore, we introduced VRR_2_ to maintain a ventilation rate sufficient to exchange the indoor air once per ∆t.

### 2.4. Field Investigation and Evaluation

This study conducted field data collection and experiments to achieve two main objectives: First, data on indoor temperature, outdoor temperature, and operation of ventilation fans and cooling pads were collected at 10 min intervals over seven rearing cycles (from 3 September 2022 to 1 September 2023, in Building 2 of the experimental farm). These data were analysed to evaluate the effectiveness of the ventilation operation and control. VRR was also calculated under the same conditions and compared with the actual ventilation rate used in the broiler house. Second, the proposed ventilation control algorithm’s performance in regulating the indoor temperature was evaluated in Building 2. Building 1 served as a control, and broilers were raised simultaneously in both buildings.

Furthermore, fewer birds than the design capacity of 30,000 were stocked to prevent losses due to heat stress during the summer, with 26,780 birds in Building 1 and 28,120 birds in Building 2. The birds were stocked on 14 August 2024, and shipped out on September 23, with early shipments of 12,000 birds from Building 1 and 13,000 from Building 2 on September 11. In addition, considering potential productivity challenges due to various stressors in the early and middle stages of growth, the proposed algorithm was applied and validated after the birds had reached stable growth, specifically from day 20 onward. Therefore, the actual application and analysis of the proposed algorithm were conducted from 12:35 p.m. on September 2 to 11:59 p.m. on 16 September 2024. During the experimental period, we compared the indoor microclimate, fan stages, fan stages and cooling pads operation, feed and water consumption, and broiler growth between the control building (Building 1) and the experimental building (Building 2).

## 3. Results and Discussion

### 3.1. Ventilation Performance of the Current Ventilation-Control System

The seven data collection periods were the following: 3 March to 3 October 2022; 5 November to 5 December 2022; 6 January to 6 February 2023; 5 March to 29 March 2023; 26 April to 25 May 2023; 14 June to 17 July 2023; and 31 July to 1 September 2023. Note that these periods do not represent the actual broiler rearing periods but rather the time frames during which valid measurement data were collected. Figure 3 illustrates the changes in indoor temperature during four of these periods, representing spring, summer, fall, and winter.

The indoor temperature gradually decreased as the broilers grew, which indicates that the temperature control was generally well maintained. However, the indoor temperatures were consistently higher during summer compared to other seasons. The difference between day and night temperatures was around 3 to 4 °C, and daytime peak temperatures were 4–5 °C higher than in other seasons, which likely caused significant heat stress to the broilers. In contrast, winter indoor temperatures were about 1 to 3 °C lower than in spring and autumn.

During the early rearing stages, seasonal differences in indoor temperature were minimal due to minimised ventilation and the need to maintain a higher optimal temperature for young chicks. However, as the broilers aged, the seasonal temperature differences became more pronounced. Notably, after day 27, indoor temperatures in winter were maintained around 23 °C, while in summer, they reached up to 30 °C, which highlights the largest seasonal discrepancy. These seasonal variations in rearing conditions will likely impede consistent productivity throughout the year. This situation highlights the need for strategies to mitigate high temperatures during the summer to ensure optimal broiler performance.

Using the rearing data collected over this year, the VRR was calculated every 10 min using the records of broiler age, indoor temperature, outdoor temperature, and temperature set points. The ventilation rate discrepancy (VR_disc_) was then determined by subtracting the actual ventilation rate from the VRR. In addition, the difference between the set and indoor temperatures was calculated and defined as the indoor temperature discrepancy (T_disc_). Figure 4 shows the indoor temperature distribution in relation to VR_disc_ and T_disc_ over the year for the ventilation operation. To better understand the impact of ventilation on indoor temperature, instances when the cooling pads were in operation were excluded.

Furthermore, a significant portion of the data fell into the quadrant with a positive VR_disc_ and negative T_disc_. These data indicate situations where the ventilation rate is lower than the VRR, which led to indoor temperatures higher than the set temperature. Conversely, scenarios in the quadrant with a negative VR_disc_ and positive T_disc_ indicate that the ventilation rate exceeded the VRR, which caused indoor temperatures to be lower than the set temperature. These scenarios suggest that if ventilation had been conducted according to the VRR, the indoor temperature might have been closer to the set temperature, which implies there is a significant potential for improvement in maintaining optimal indoor conditions.

While situations with both positive VR_disc_ and T_disc_ were rare, a considerable amount of data fell into the quadrant with both negative VR_disc_ and T_disc_. This indicates that in cases where the ventilation rate was above the VRR, the indoor temperatures were higher than the set temperature. These situations contradicted the intended purpose of the VRR and appeared scattered without a clear trend or specific pattern. These occurrences were likely due to time lags in the effect of ventilation on indoor temperature, as temperature adjustments do not occur instantaneously after ventilation changes.

The proportions of data points in each quadrant are shown in Figure 5. By defining the proper indoor temperature as being within ±1.5 °C of the set temperature, it was determined that 74% of the total data collection time maintained the proper rearing temperature. Conversely, for 26% of the total rearing time, the indoor temperature was not maintained within the desired range. Among these, 15% of the time, high temperatures were observed due to insufficient ventilation, while 3% of the time, low temperatures occurred due to excessive ventilation. This suggests that if ventilation had been adjusted according to the VRR, proper temperatures could have been maintained for an additional 18% of the time, potentially achieving optimal rearing conditions for up to 92% of the total rearing period. More studies are needed to understand the physical mechanisms behind the remaining 8% of cases where both over-ventilation and high temperatures occurred simultaneously.

### 3.2. Ventilation Performance of the Proposed Ventilation Control System

During the experimental period, comparisons were made between the control building (Building 1) and the experimental building (Building 2). This was carried out to evaluate the indoor microclimate conditions, fan stages, operation of fan stages and cooling pads, feed and water consumption, and overall growth performance. The period from day 26 to day 28 was excluded from the comparison due to algorithm maintenance, and day 29 was excluded because some broilers were harvested early.

First, the similarity between the control and experimental buildings was assessed. Figure 6 shows the indoor temperature changes up to day 20, during which both buildings operated under the existing ventilation control algorithm. The indoor temperatures in both buildings followed nearly identical patterns across different ages and between day and night. This confirmed that it was suitable to evaluate the effects of the new ventilation control system in the two buildings.

As shown in Figure 7, daytime relative humidity was almost identical in both the control and experimental buildings. However, at night, the relative humidity in the control building was about 5% higher than in the experimental building. During the day, high ventilation rates resulted in active air exchange, which caused indoor relative humidity to be strongly affected by the outdoor conditions. In contrast, at night, reduced ventilation rates led to indoor humidity being more affected by internal moisture dynamics. Consequently, Building 1 showed approximately 5% higher relative humidity at night than Building 2—likely due to internal moisture mechanisms that are not fully understood. This difference should be taken into account in comparative experiments.

#### 3.2.1. Evaluating the Indoor Microclimate Conditions

During the experimental period, extremely hot weather was observed, with outdoor temperatures ranging from 33 °C to 37 °C during the day (Figure 8). Even at night, outdoor temperatures often exceeded 26 °C, which was higher than the set temperature of 24–26 °C and created challenging conditions for microclimate control. Ideally, cooling systems would be used to maintain the set temperature even at night, but practical considerations such as energy costs and the ability of grown broilers to tolerate higher temperatures led to the decision to rear the broilers at temperatures above the set point. Figure 8 shows that neither building could reach the set temperature indoors. When comparing the indoor temperatures of the experimental and control buildings, it was found that the experimental building with the proposed ventilation control algorithm maintained temperatures 1.5–2 °C lower than the control building during the day and up to 1.5 °C lower or similar at night. On certain occasions, the indoor temperature in the experimental building exceeded that of the control building. Specifically, during three periods—day 26 from 12:06 p.m. to 4:34 p.m., day 27 from 11:00 a.m. to 12:40 p.m., and day 28 from 11:24 a.m. to 5:43 p.m.—the ventilation control system was temporarily reverted to the conventional system for maintenance purposes, resulting in suboptimal control of the indoor temperature.

Compared to the experimental building, it was obvious that the proposed ventilation control algorithm maintained indoor temperatures closer to the optimal range for broilers. However, to further improve the performance of the algorithm, it is necessary to address its current limitations. First, the energy balance model in Equation (1) assumes complete mixing of the indoor air, which requires correction. Considering the localized temperature variations caused by ventilation can lead to a better algorithm. Second, although the theoretical cooling effect of cooling pads is quite high, in practice, their effectiveness varies depending on the saturation level of the pads. There are also differences in localized cooling effects within the cooling pad itself. Accounting for these practical aspects of cooling pad performance can help improve the algorithm’s overall effectiveness.

During the day, outdoor relative humidity ranged from 45–60%, while at night, it increased to 85–95%, as shown in Figure 9. Indoor relative humidity during the day ranged from 60–75%, which was higher compared to the outdoor levels. At night, indoor relative humidity reached 80–95%. The consistently high indoor humidity can be attributed to the broilers and their activity, which continuously added moisture to the environment. Ventilation during the day helped reduce humidity levels, while at night, reduced ventilation resulted in lower indoor relative humidity compared to outside. When comparing the control and experimental buildings, the indoor relative humidity in the experimental building was slightly higher than in the control building during the day, with a difference below 5%. At night, however, the relative humidity in the control building was about 5% higher than in the experimental building, which was consistent with the trend observed before day 20—see Figure 7. In other words, the proposed ventilation control algorithm resulted in slightly higher (up to 5%) indoor relative humidity during the day but did not affect nighttime humidity. This increase in daytime relative humidity can be attributed to the operation of the cooling pads, as discussed in Section 3.2.2.

#### 3.2.2. Evaluation of the Operation of Tunnel Fans and Cooling Pads

Figure 10 and Figure 11 compare the use of tunnel fans and cooling pads during the experimental period. During the day, the control building consistently reached the fan stage, which suggests that all 14 tunnel fans were operating. In contrast, the experimental building operated with fewer tunnel fans—typically at fan stages 3 to 4. Both buildings operated at fan stages 1–2 at night, but the experimental building consistently maintained a lower fan stage.

Moreover, the use of the cooling pads was significantly higher in the experimental building than in the control building. In the experimental building, the cooling pad operated almost continuously during the day, while in the control building, it ran intermittently for 40 to 80 s every five minutes. However, during days 33 and 34, there were periods when cooling pad usage in the control building was relatively high. Note that excessive use of the cooling pads can potentially increase indoor humidity, but as shown in Figure 9, relative humidity only rose by about 5% (at most), which suggests minimal impact.

Figure 12 shows the relationship between outdoor temperature and ventilation rate in the experimental and control buildings. In the control building, which used the existing system, the ventilation rate increased stepwise with outdoor temperature, indicating that the operation of the tunnel fans was significantly dependent on the outdoor temperature. The cooling pads also operated only when the outdoor temperature exceeded 30 °C, and when they were active, all tunnel fans were operating at maximum capacity. In contrast, the experimental building showed that the ventilation rate was not solely determined by the outdoor temperature. The distribution of ventilation rates was distinctly different from that of the control building, as factors such as broiler age and indoor temperature influenced the decision on ventilation rate. Additionally, the cooling pads were active in most cases as the outdoor temperature increased. However, when the cooling pads were in operation, the ventilation rate varied flexibly from low to high levels. In some cases, the ventilation rate was as low as 148,000 m^3^ h^−1^ when the outdoor temperature exceeded 31 °C. This occurred after the cooling pads had lowered the indoor temperature, and the system maintained a low ventilation rate for a few minutes to stabilise the indoor temperature. In the control building, the conventional ventilation system was operated at fan stage 1, a ventilation rate of 148,000 m^3^ h^−1^, or above because the experiment was conducted during summer and after day 20. In contrast, the experimental building was controlled based on the energy balance calculation, allowing the fan stage to reach 0 state at some time points, which resulted in ventilation rates lower than 148,000 m^3^ h^−1^.

The total usage of tunnel fans during the experimental period was 3182 fan hours in the control building and 1645 fan hours in the experimental building. This indicates that the experimental building with the proposed ventilation control system reduced the tunnel fan usage to 51.69%. On the other hand, the cooling pad usage was 11.75 h and 88.80 h for the control and experimental buildings, respectively. This indicates that the cooling pad usage in the experimental building increased to 7.56 times.

Note that the reduction in tunnel fan usage with the proposed ventilation control system has significant implications for energy consumption. Each tunnel fan consumes 836 W of electricity, resulting in energy usage of 2660 kWh in the control building and 1375 kWh in the experimental building during the experimental period. This suggests that approximately 1285 kWh of energy was saved in the experimental building. Although the cooling pad usage increased to 7.56 times, the electricity consumption by the cooling pads was 12.9 kWh in the control building and 97.7 kWh in the experimental building, which led to an increase of only 84.75 kWh in the experimental building. While the amount of water used by the cooling pads increased significantly, the experimental farm uses groundwater, so the impact on cost was minimal. However, implementing measures such as a water circulation system will be essential in the future to conserve water resources.

#### 3.2.3. Evaluation of the Growth Performance

The measurement of the feed and water consumption for individual chickens was challenging; thus, it was estimated indirectly by measuring the daily amounts supplied to each building. Figure 13 shows the daily feed and water supply per bird, which was calculated by dividing the total daily supply by the number of chickens in each building. Generally, water consumption in chickens is about twice the feed intake, a trend observed throughout the study. The feed supply in both the control and experimental buildings was similar, with slightly higher levels in the control building. Water supply, however, was substantially higher in the control building, likely because the higher indoor temperatures caused increased water intake in response to heat stress. Following the early shipment on day 29, the feed and water supply showed slight instability. As shown in Figure 13, the average weight of the chickens in both buildings did not differ significantly, which means that productivity was not significantly affected.

Figure 14 shows that cumulative death and culling rates from day 20 to day 29 were approximately 16.5% lower in the experimental building compared to the control building. Despite the higher number of chickens in the experimental building, the lower mortality rate can be attributed to the introduced ventilation control system, which helped maintain lower indoor temperatures and reduce heat stress. However, after the early shipment on day 29, the mortality rate in the experimental building began to increase significantly starting from day 31. Our analysis revealed that a respiratory disease caused this increase in mortality in certain areas within the experimental broiler house. This conclusion is also reflected in Figure 13, where a significant drop in feed supply in the experimental building was observed on day 32.

In summary, the proposed ventilation control system effectively reduced heat stress in the experimental building compared to the control building, which led to a lower overall mortality rate. Nevertheless, due to the limited duration of system implementation, it was impossible to ascertain whether the system contributed to an increase in average chicken weight. Future studies with more extended observation periods are needed to fully evaluate the impact of the new system on growth performance and productivity metrics.

## 4. Conclusions

This study introduced a novel ventilation control algorithm that determines the required ventilation rate through heat energy balance analysis and evaluated its effectiveness in controlling the microclimate within a commercial broiler house. Initially, operational data from the experimental farm were analysed, where the existing ventilation control system was in use for over a year, including seven distinct rearing periods. The analysis of the ventilation performance showed that, for 74% of the entire study period, indoor temperatures were maintained within ±1.5 °C of the set temperature, while for 26% of the time, temperatures deviated from the target range. The investigation revealed that proper adjustment of ventilation rates based on the proposed algorithm could enhance microclimate conditions by up to 18% during periods of deviation, potentially achieving optimal conditions for as much as 92% of the entire rearing period.

The proposed algorithm was implemented and evaluated at two broiler houses (control and experimental) on the experimental farm starting from day 20 to day 34 during a rearing period under high-temperature conditions. The findings from this study indicate that the proposed algorithm could significantly improve the microclimate in a mechanically ventilated broiler house by reducing heat stress, especially during high-temperature periods. The experimental building showed a reduction in indoor temperatures of up to 2 °C compared to the control building and a decrease in tunnel fan usage to approximately 51.69%. This enabled substantial electric energy savings. Even though the cooling pad usage increased to 7.56 times, the resulting additional energy consumption was relatively small, and the overall benefits of mitigating heat stress clearly outweighed the costs.

In addition, the implementation of the proposed ventilation algorithm was associated with a 16.5% reduction in overall mortality compared to the control building. This outcome suggests significant improvements in bird rearing conditions as well as (potentially) enhanced productivity. Unfortunately, the relatively short application period of the algorithm hindered the ability to draw definitive conclusions regarding its impact on average chicken weight. However, it is evident that improvements to the ventilation algorithm can enable chickens to grow in optimal rearing conditions for extended periods, particularly by reducing their exposure to high-temperature environments. These findings underscore the necessity for longer-term studies to comprehensively assess the system’s potential benefits for productivity, particularly in relation to growth performance. Ultimately, successful implementation of the system can help minimise operator intervention, create optimal rearing conditions for broilers, and enhance production efficiency.

## Figures and Tables

**Figure 1 animals-14-03430-f001:**
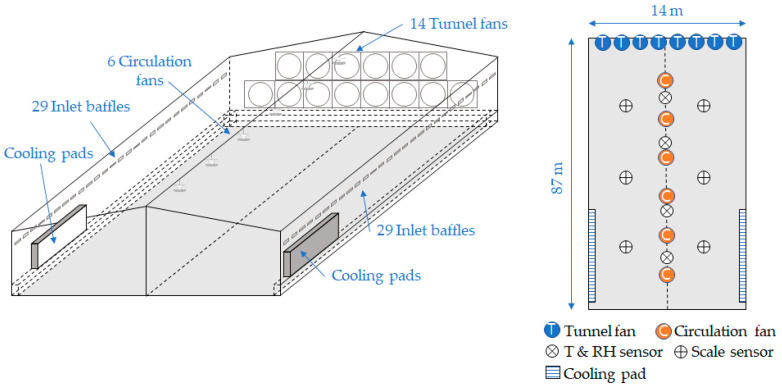
Schematic of the experimental broiler house and measurement layout. The figure was modified from the drawing by Choi et al. [21].

**Figure 2 animals-14-03430-f002:**
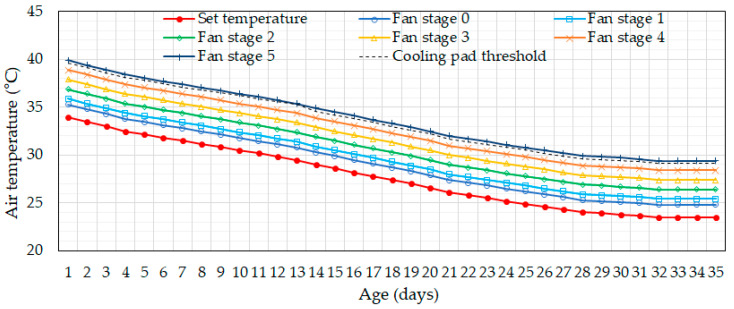
Set temperature values for six different fan stages and cooling pad thresholds for different chicken ages were used for the current ventilation control on the experimental farm.

**Figure 3 animals-14-03430-f003:**
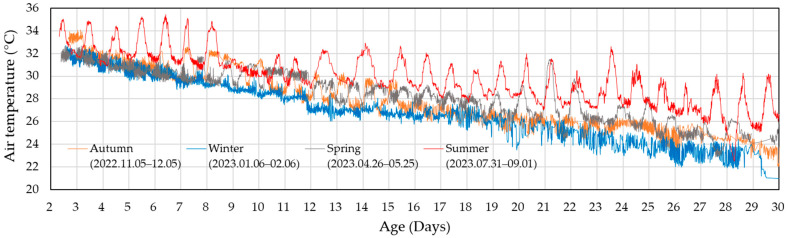
Changes in indoor temperature during data collection periods, representing the seasons of spring, summer, autumn, and winter.

**Figure 4 animals-14-03430-f004:**
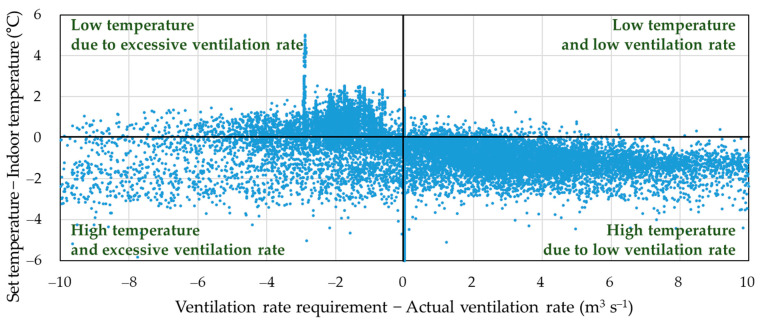
Distribution of ventilation rate discrepancy (ventilation rate requirement minus actual ventilation rate) and indoor temperature discrepancy (set temperature minus indoor temperature).

**Figure 5 animals-14-03430-f005:**
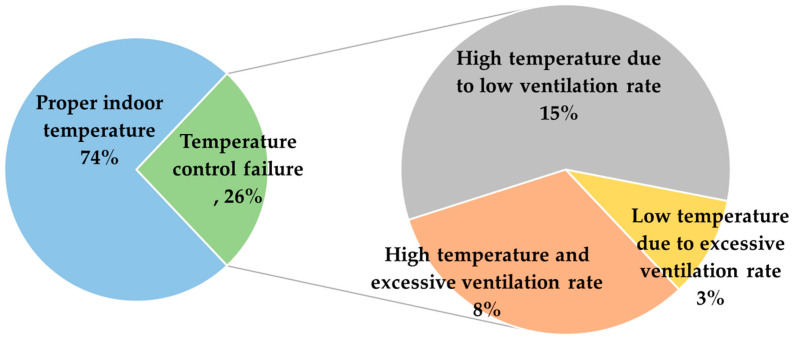
Comparison of ventilation rate and resulting indoor temperature at Farm A over one and a half years against the ventilation rate requirement.

**Figure 6 animals-14-03430-f006:**
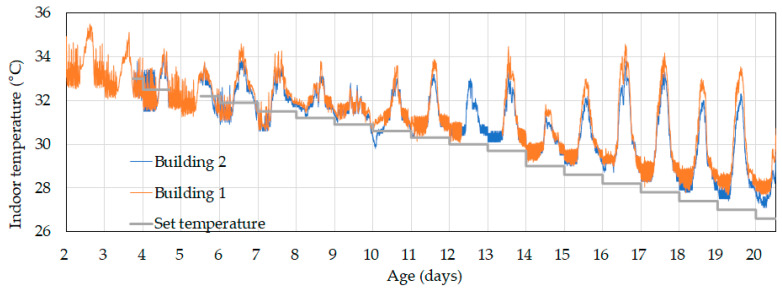
Indoor temperature changes in the control (Building 1) and experimental (Building 2) buildings until day 20 using the existing ventilation control algorithm. There are missing data from Building 1 on days 12 and 13 and from Building 2 on days 2 to 5.

**Figure 7 animals-14-03430-f007:**
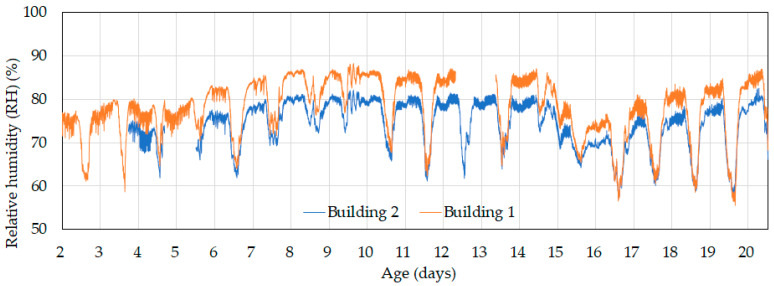
Relative humidity changes in the control (Building 1) and experimental (Building 2) buildings until day 20 using the existing ventilation control algorithm. There are missing data from Building 1 on days 12 and 13 and from Building 2 on days 2–5.

**Figure 8 animals-14-03430-f008:**
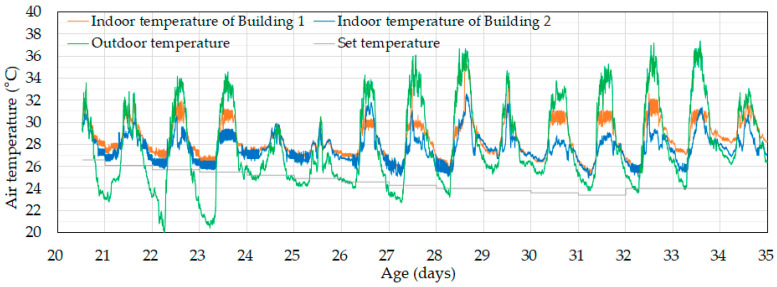
Changes in indoor temperature in the control (Building 1) and experimental (Building 2) buildings starting from day 20.

**Figure 9 animals-14-03430-f009:**
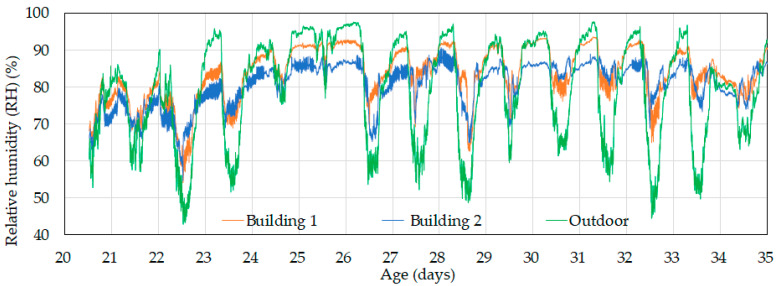
Changes in indoor relative humidity in the control (Building 1) and experimental (Building 2) buildings starting from day 20.

**Figure 10 animals-14-03430-f010:**
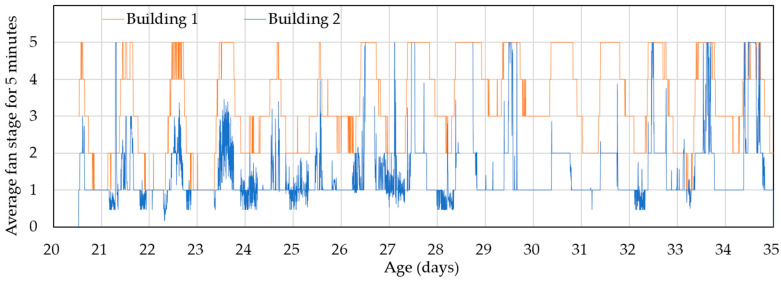
Changes of the fan stage in the control (Building 1) and experimental (Building 2) buildings starting from day 20. The fan stages are represented as average values calculated over 5 min intervals.

**Figure 11 animals-14-03430-f011:**
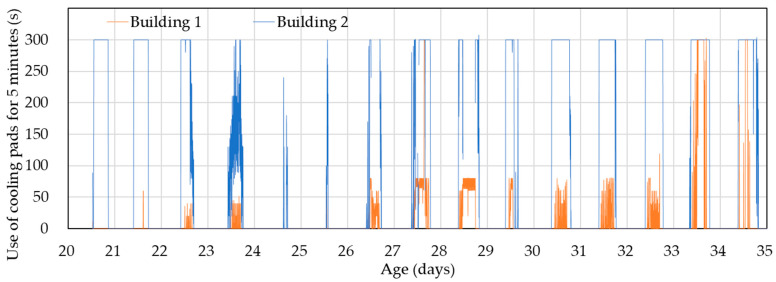
Changes in cooling pad usage for the control (Building 1) and experimental (Building 2) buildings starting from day 20.

**Figure 12 animals-14-03430-f012:**
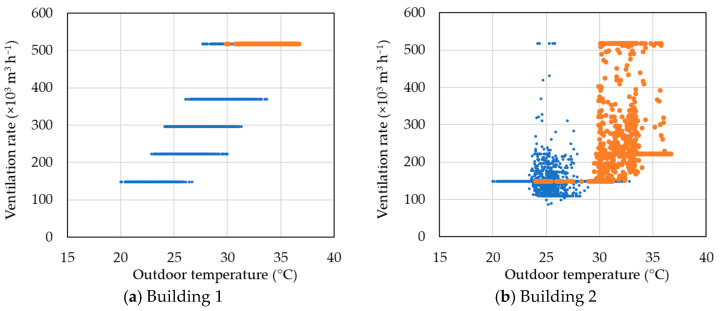
Scatterplot of the relationship between outdoor temperature and ventilation rate as average values calculated over 5 min intervals. Blue points represent cases when the cooling pad was not operating, while orange points represent cases when the cooling pad was operating.

**Figure 13 animals-14-03430-f013:**
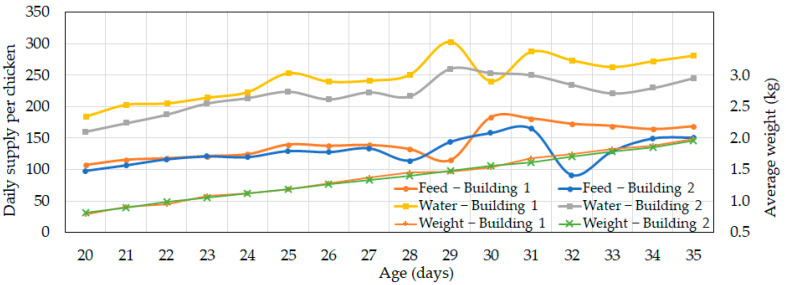
Changes in daily feed and water supply per bird and predicted average chicken weight in the control (Building 1) and experimental (Building 2) buildings starting from day 20 (feed supply shown in g chicken^−1^ day^−1^, water supply shown in mL chicken^−1^ day^−1^).

**Figure 14 animals-14-03430-f014:**
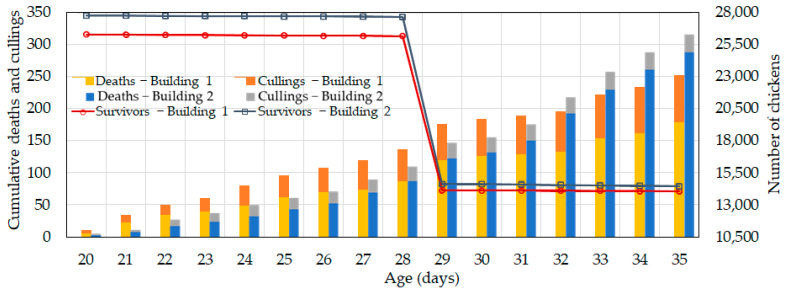
Cumulative number of deaths, cullings, and survivors in the control (Building 1) and experimental (Building 2) buildings starting from day 20.

## Data Availability

The data presented in this study are available on request from the corresponding author. The data are not publicly available due to privacy concerns.

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
