# Peer review of "Preliminary Evaluation of an Advanced Ventilation-Control Algorithm to Optimise Microclimate in a Commercial Broiler House"

_animals, 2024, doi:10.3390/ani14233430_

Round 1

Reviewer 1 Report

Comments and Suggestions for Authors

Review of Manuscript: "Evaluation of an Advanced Ventilation-Control Algorithm to Optimise Microclimate in a Commercial Broiler House"

General Comments:

This manuscript explores the development and evaluation of a ventilation control algorithm aimed at enhancing the microclimate in commercial broiler houses. The topic is of considerable relevance given the global importance of efficient livestock farming and animal welfare. While the manuscript provides comprehensive insights into the proposed algorithm's benefits, including reductions in heat stress, energy consumption, and bird mortality, certain aspects require clarification and improvement to enhance the manuscript's quality and scientific robustness.

Suggestions for Improvement:

1. Building System Parameters:

   - Present detailed specifications for the experimental and control buildings, including:

     - Minimum and maximum ventilation rates (based on COâ‚‚ thresholds and heat removal requirements).

     - A graphical representation of ventilation rate relationships to indoor temperature.

   - Specify the set-point temperature and analyze the volume flow in relation to the COâ‚‚ production of the broilers. This will enable better evaluation of air quality and whether the ventilation system meets the animals' physiological needs.

2. Thermal Analysis:

   - Use heat stress parameters, such as the number of hours per year exceeding certain thresholds, to quantitatively assess the thermal conditions.

   - Address how the reduction in temperature (e.g., 1.5–2 °C) impacts broiler welfare and productivity in detail.

3. Data Presentation:

   - Reorganize data visualization:

     - Include graphs showing volume flow rate as a function of indoor temperature and outdoor conditions.

     - Differentiate data points where cooling pads were active using visual markers (e.g., colors).

     - Replace relative humidity metrics with absolute humidity (e.g., water vapor density) for a clearer interpretation of the building's moisture dynamics. Use psychrometric charts (h,x diagrams) to illustrate these changes.

4. Cooling Pad Contribution:

   - The role of cooling pads in the observed outcomes is not discussed adequately. Separate the algorithm's effect from the contribution of cooling pads. Quantify their relative impacts on temperature reduction and energy consumption.

5. Comparative Analysis:

   - Include a comparative discussion with existing ventilation control strategies, particularly those using cooling pads, across different livestock species (e.g., swine). This would contextualize the findings and highlight the algorithm's novelty.

6. Discussion Depth:

   - Expand the discussion to consider studies addressing:

     - Advanced control strategies for confined livestock environments.

     - Long-term implications for growth performance and energy use.

     - Feasibility of implementation at large-scale farms.

7. Minor Remarks:

   - Use Kelvin (K) instead of Celsius (°C) for temperature differences (e.g., thermal transmittance calculations).

   - Incorporate metrics like daily feed intake, feed conversion ratios, and water usage into the performance analysis for a more holistic evaluation.

Language and Style:

While the manuscript is well-written, a few grammatical and stylistic errors detract from its clarity:

   - Replace "a optimum" with "an optimum."

   - Avoid redundancy, such as "For the this paper."

   - Proofread the text to correct typos like "livestovk" and "presents" instead of "presented."

Conclusion:

The study presents a valuable contribution to improving broiler house management. By addressing the above recommendations, the manuscript would significantly enhance its scientific rigor, practical applicability, and alignment with the journal's scope.

Author Response

Thank you for your questions and comments. We have prepared a response document with point-by-point answers to all comments, and please refer to the attached file.

Reviewer 2 Report

Comments and Suggestions for Authors

Once it was reviewed the article, I found that is very appropriate and it will contribute to this knowledge field, Nevertheless, I have some recommendations and suggestions:

1. In the simple summary and the abstract the value of mortality are not the same (16,5% and 17%), It would be important  to use the same value.

2. In Materials and methods in "2.3 ventilation control algorithms", It is important to explain better the current ventilation control, which the authors have referred as an empirical system control to operate tunnel fans and cooling fans, in order to determinate which variables are taking into account in this system.

3. Figure 4, could be explained better in order to the reader could understand the results and the needs of improve the current empirical ventilation system

4. Is it possible to improve the Figure 5?

5. It was assessment the algorithm that was proposed to ventilation control system, and the results that are shows in the figures 10-13, that’s why it is possible to explain more if there were significant different statistically between building 1 and 2, isn´t it? Could you do that?

Author Response

(The authors gave the same response as above.)

Reviewer 3 Report

Comments and Suggestions for Authors

This study presents an improved ventilation algorithm to optimize microclimate conditions in broiler houses. It has the potential to improve efficiency for broiler houses. One year of comprehensive data was collected and analyzed to demonstrate the algorithm's effectiveness, which is important to increase the study's credibility. A 17% mortality reduction and improvements in energy savings to ensure both energy savings and animal welfare indicate that the application can be effective. In addition, testing the algorithm of the study in a commercial broiler farm provided a concrete and practical perspective on the application. However, in addition to these, some suggestions that should be made in the study are listed below.
.  The implementation of the algorithm and the increase in the use of cooling pads in broiler houses have not been evaluated, especially without cost analysis.
. Regarding applicability to farms, installation costs and sustainability analyses need to be evaluated.
. The study only analyzed the short-term effects of the algorithm's performance.  However, no clear information was provided on how it could have an impact on the growth process of the chickens in the long term.
. It should be evaluated more thoroughly with long-term observations. The differences between control and experimental buildings should have been supported by a more comprehensive analysis. For example, not only temperature but also other environmental factors, such as humidity, ammonia, etc., could be examined in more detail in terms of their impact on animal welfare. The growth performance of chickens should be studied in more detail.  More data on parameters such as feed consumption and weight gain would be helpful to evaluate the direct impact of the algorithm on animal productivity.
. The climatic characteristics of the location where the study was conducted, limiting how this method can be applied in other geographical regions, should be discussed further in relation to the suitability or adaptability of the algorithm to different climatic conditions.
. Whether the algorithm used has been optimized through modeling or simulation should be indicated. Besides modeling, discussing various scenarios to improve the algorithm's performance would be good.

It is a study that will make a useful and innovative contribution to broiler chickens' welfare and energy efficiency.  However, long-term impact analyses, cost assessments, and further investigation of its suitability for different climatic conditions are critical for the results to be generalizable and widely adopted in the agricultural sector.  Once these shortcomings are addressed, the study will be a good study with the potential for broader application.

Author Response

(The authors gave the same response as above.)

Reviewer 4 Report

Comments and Suggestions for Authors

Review: Evaluation of an Advanced Ventilation-Control Algorithm to Optimise Microclimate in a Commercial Broiler House

I evaluated the manuscript to ensure readiness in terms of innovation, completeness, and clarity. Please find my thoughts below.

1. Innovation

Originality: The study introduces a novel ventilation control algorithm based on a heat-energy balance approach to optimize microclimates in broiler houses. The algorithm's integration of dynamic ventilation needs, accounting for indoor and outdoor conditions, provides an innovative solution compared to traditional static temperature-based controls.

Advancement Over Current Techniques: The manuscript effectively indicates the new algorithm's benefits over existing ventilation controls, especially in reducing heat stress and conserving energy. However, further emphasizing how this model compares with alternative state-of-the-art systems (beyond static temperature methods) could reinforce its unique contributions.

Broader Applications: Extending the discussion of how this algorithm could potentially be adapted for various climates or different poultry species would enhance its relevance and applicability.

2. Completeness

Methods and Experimental Design: The methodology is thorough, detailing the experimental setup, control parameters, and comparison metrics between the two buildings. Nevertheless, additional information on the algorithm's computational complexity or operational ease could improve understanding of its practical implications for farm operators.

Data and Statistical Analysis: The manuscript effectively uses ventilation rate discrepancy and temperature discrepancy as metrics to evaluate system performance. However, statistical validation of the observed differences in performance metrics (temperature, humidity, mortality rates) between experimental and control settings could encourage scientific rigor.

Potential Limitations: While the manuscript acknowledges that further studies are required to assess long-term productivity impacts, it could benefit from a more detailed discussion on limitations, such as operational challenges or resource requirements in implementing the algorithm across different farm setups.

3. Clarity

Structure and Flow: The manuscript is well-organized, progressing logically from problem statement to methodology, results, and conclusions. Using figures (e.g., schematics of fan stages and temperature discrepancies) is adequate, though more concise captions would aid reader comprehension.

Technical Language: While the manuscript maintains an academic tone, some technical terms (e.g., "VRR" for ventilation rate requirement) could be briefly defined upon the first usage.

Abstract and Conclusion: The abstract clearly summarizes the study's objectives and outcomes. However, the conclusion could be drawn up by explicitly stating the practical significance of the results, particularly for farmers and industry stakeholders, making the work more impactful.

Areas for Improvement:

1. Comparison with Alternative Methods: A brief comparison with other advanced ventilation systems in commercial use (e.g., sensor-driven adaptive ventilation systems) could add better context.

2. Increased Statistical Validation: Applying statistical significance testing to support performance improvements (e.g., mortality reduction, energy savings) would strengthen the results section.

3. Highlighting Broader Implications: Expanding on how the study's findings can be applied to different environmental conditions or poultry farming setups could enhance the manuscript's appeal to a broader audience.

Author Response

(The authors gave the same response as above.)

Round 2

Reviewer 1 Report

Comments and Suggestions for Authors

You decline nearly all my suggestions in an overly verbose manner. I will illustrate this with some examples:

The minimum ventilation rate is calculated by dividing the COâ‚‚ release of the animals by the maximum allowable difference between indoor and outdoor COâ‚‚ concentrations. For instance, if the maximum accepted indoor concentration is 2000 ppm, the required volume flow rate can be determined accordingly.

The maximum ventilation rate can be calculated based on the sensible heat release of the animals and the heat capacity of the air (depending on the temperature) for both the inlet and outlet air. The maximum allowable difference between indoor and outdoor temperatures is a crucial parameter for calculating the ventilation flow rate needed to prevent excessive indoor temperatures.

The measured ventilation flow rate (Y) can be visualized using scattergrams as a function of outdoor temperature (X) and indoor temperature (X). Additionally, the indoor temperature (Y) can be plotted against outdoor temperature (X). Comparing these metrics across the two buildings would reveal distinct characteristics of the respective ventilation systems.

Please explain in detail how the indoor temperature can be reduced when the ventilation flow rate is lowered. How do you address the removal of the animals' sensible heat under such conditions?

You argued that heat stress parameters are inapplicable because broilers are harvested before 35 days. This claim fails to acknowledge that even short-term exposure to high temperatures can have cumulative effects on productivity and welfare. Dismissing this suggestion without addressing these impacts limits the study's practical relevance.

Author Response

Thank you for your questions and comments. We have made a response document with point-by-point answers to all comments, and please refer to the attached file.

Round 3

Reviewer 1 Report

Comments and Suggestions for Authors  

Here is the combined and improved version of your text:

In your response, you have once again chosen to address only parts of my suggestions while disregarding others, often without sufficient justification. This piecemeal approach to incorporating feedback is not acceptable and significantly diminishes the overall quality and rigor of the manuscript. I must emphasize that addressing comments partially or selectively does not meet the standard expected for a thorough review process.

In general, I cannot imagine that meaningful improvements to the manuscript can be implemented soundly within 24 hours, as suggested by the timing of your responses. Your strategy of picking up only a few of my suggestions while rejecting others—often without adequate explanation—is evident.

For instance, in this latest revision, you have addressed only one suggestion regarding the graphical presentation of the measurements, adding a single figure. While this addition is a step in the right direction, it remains insufficient. I strongly recommend adding more figures in the proposed manner. The additional figure (ventilation flow rate vs. outdoor temperature) supports the validity of my concerns. Even at high outdoor temperatures (above 30°C), the ventilation rate remains surprisingly low. Please provide a detailed explanation of how the indoor temperature can be reduced by approximately 2 K, even when the ventilation rate is lower compared to the reference building.

Moreover, I recommend incorporating the minimum and maximum ventilation rates into Figure 18 for clarity. There is a glaring discrepancy between your calculations and the data presented in Figure 18: while you state the minimum ventilation flow rate is between 1700 m³/h and 22000 m³/h, the figure shows a minimum rate closer to 150000 m³/h—a tenfold difference. This discrepancy must be explained thoroughly, as it raises significant questions about the accuracy and consistency of your findings.

Once again, I strongly urge you to revisit and fully address the suggestions from my initial review. As demonstrated in Figure 18, the proposed presentation of the data provides critical insights into the interaction between the livestock building and the ventilation system. Additionally, I recommend taking the necessary time to carefully improve the manuscript rather than submitting incomplete revisions. Selective and superficial responses will not suffice for the next iteration, and a more committed effort is essential to bring the manuscript to a publishable standard.

Author Response

Thank you for your comments and suggestions. Please find the attachment for our response.
